# Structural basis for substrate specificity and regulation of nucleotide sugar transporters in the lipid bilayer

Joanne L. Parker [1]*, Robin A. Corey[1], Phillip J. Stansfeld [2]* & Simon Newstead [1]*

Nucleotide sugars are the activated form of monosaccharides used by glycosyltransferases during glycosylation. In eukaryotes the SLC35 family of solute carriers are responsible for their selective uptake into the Endoplasmic Reticulum or Golgi apparatus. The structure of the yeast GDP-mannose transporter, Vrg4, revealed a requirement for short chain lipids and a marked difference in transport rate between the nucleotide sugar and nucleoside monophosphate, suggesting a complex network of regulatory elements control transport into these organelles. Here we report the crystal structure of the GMP bound complex of Vrg4, revealing the molecular basis for GMP recognition and transport. Molecular dynamics, combined with biochemical analysis, reveal a lipid mediated dimer interface and mechanism for coordinating structural rearrangements during transport. Together these results provide further insight into how SLC35 family transporters function within the secretory pathway and sheds light onto the role that membrane lipids play in regulating transport across the membrane.

---

[1] Department of Biochemistry, University of Oxford, Oxford OX1 3QU, UK. [2] School of Life Sciences & Department of Chemistry, The University of Warwick, Coventry CV4 7AL, UK. *email: joanne.parker@bioch.ox.ac.uk; phillip.stansfeld@warwick.ac.uk; simon.newstead@bioch.ox.ac.uk

Glycosylation plays a crucial role in cell physiology, impacting numerous process from quality control during protein folding and subsequent protein trafficking, to cell recognition, developmental signalling and immune system function[1,2]. Changes in cell surface glycosylation can modulate inflammatory responses, promote cancer cell metastasis and regulate apoptosis. Understanding the pathways that control glycosylation in eukaryotic cells is therefore at the forefront of the emerging field of glycomedicine, which promises new therapeutic routes to many human diseases[3].

In eukaryotes, glycosylation occurs in the lumen of either the Endoplasmic Reticulum (ER) or Golgi Apparatus (Golgi), where specific glycosyltransferase enzymes recognise donor sites on either proteins or lipids to build up a complex array of carbohydrate structures[4]. However, to function efficiently, glycosyltransferases require a continuous supply of activated nucleotide sugars, which act as the precursors to the glycan structures that are built on protein and lipid molecules. Nucleotide sugars contain a single carbohydrate attached via a phosphodiester bond to either a nucleoside diphosphate or nucleoside monophosphate carrier group[5]. The labile nature of the phosphate bond plays an important role in the enzymatic mechanism that transfers the sugar group during glycosylation. Mammals use nine nucleotide sugars for glycosylation[5], with CMP-Sialic acid and GDP-fucose being the only ones to utilise cytosine and guanine nucleotides, whilst the remaining seven sugars are all conjugated to uridine.

Nucleotide sugars are synthesised in either the cytoplasm or nucleus and must be transported across the ER and Golgi membranes through the action of nucleotide sugar transporters (NSTs)[6]. Following their use by the glycosyltransferases the resulting nucleoside diphosphate is then processed by a luminal phosphatase, before being transported back across the membrane as a nucleoside monophosphate (NMP) through the same transport proteins[7]. NSTs thus play an important role in regulating glycosylation by controlling the transport of essential activated sugars to the glycosyltransferase enzymes and transporting the NMPs back into the cytosol to be regenerated.

NSTs belong to the SLC35 family of solute carriers and are highly conserved from simple eukaryotes like fungi and parasites to complex multicellular organisms such as plants and mammals[8]. NSTs are divided into seven subfamilies, designated A-G based on the specific nucleotide sugar they transport[9]. The first crystal structure for a member of the SLC35 family, the GDP-mannose transporter from *Saccharomyces cerevisiae* Vrg4, was recently reported in its substrate free and nucleotide sugar bound states[10]. This was recently followed by structures of both the mouse and maize CMP-sialic acid transporters[11,12]. These structures reveal a conserved architecture for the SLC35 family consisting of 10 transmembrane helices arranged around a central ligand binding site in a five plus five configuration[13].

The transport of GDP-mannose is important for pathogenic fungi, such as *Candida albicans*, *Aspergillus fumigatus* and *Cryptococcus neoformans*[14,15]. These organisms contain a cell wall predominantly formed of glycomannosylated conjugates that form a protective coat against the human immune system. GDP-mannose transport is fundamental for virulence and essential for cell survival, making the transporters attractive targets for inhibitor design[16]. The recent crystal structure of Vrg4 bound to GDP-mannose (PDB: 5OGE) now provides opportunities for structure-based drug design targeted at this class of transporter. The structure revealed specificity pockets that separate recognition of the nucleotide and sugar groups, revealing sequence motifs that appear to correlate with substrate specificity. Nevertheless, structural information concerning the recognition of GMP still remains unknown, hampering efforts to fully understand ligand recognition. It was also observed that transport rates between GDP-mannose and the anti-ported ligand, guanine monophosphate (GMP), were different; with GDP-mannose being transported at a faster rate compared to GMP[10]. This difference in transport rate may explain how NSTs achieve directional transport across the ER and Golgi membranes. The

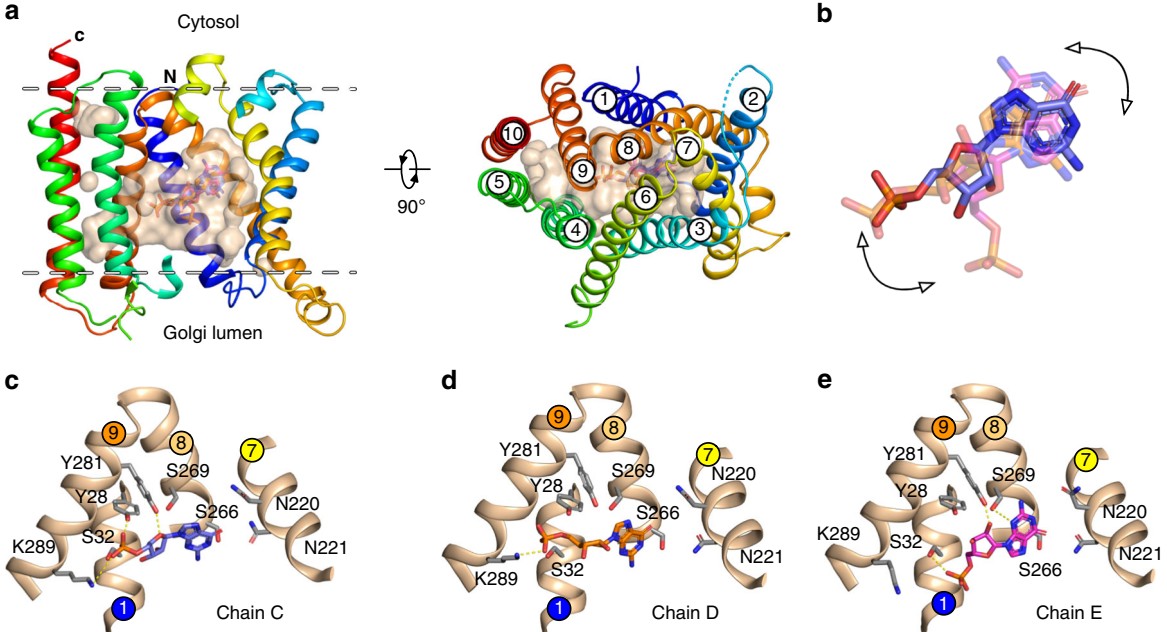

**Fig. 1** Crystal structure of Vrg4 bound to GMP. **a** Crystal structure of Vrg4 with helices coloured blue (N-terminus) to red (C-terminus). The binding cavity is open to the Golgi lumen, and shown as a transparent surface (wheat). The three different positions observed for the GMP molecule are overlaid in blue, orange and pink. The position of the lipid bilayer is indicated by white dashed lines. **b** An overlay showing the three GMP molecules as presented in **a**, with arrows highlighting the changes in positions observed for the nucleoside and phosphate groups. **c**, **d**, **e** Zoomed in views of the binding site from chains C, D and E respectively, showing the different positions and interactions made to GMP in the crystal structure

discovery that Vrg4 requires short chain lipids to function also suggested a second route by which nucleotide sugar transport could be regulated in the cell. Given the highly dynamic membrane environment of the ER and Golgi, the concept that transport proteins can be regulated using specific lipids and bulk bilayer properties, such as local structure and membrane thickness, is an exciting, but poorly understood area of membrane biology.

To address these questions we sought to determine the structure of Vrg4 bound to the nucleoside monophosphate, GMP and analyse how lipids regulate the oligomeric state and function of the transporter. Our structure reveals that GMP is accommodated in different orientations within the binding site, explaining why counter transport of this ligand is less efficient than with GDP-mannose. Using a combination of coarse grained and all atom molecular dynamics, our study also demonstrates how short chain lipids likely regulate transporter function and facilitate dimer formation in the membrane.

## Results

**GMP adopts different conformations in the binding site**. The difference observed between the transport of GDP-mannose and GMP raised the question of whether the nucleoside monophosphate was recognised in a different way within the binding site. To understand how GMP is recognised by Vrg4 we determined a co-crystal structure with GMP at 3.3 Å resolution (Fig. 1a and Supplementary Table 1). Vrg4 adopts a very similar pose as that for the GDP-mannose transporter (root mean square deviation ~0.47 Å between 276 $C_\alpha$ atoms), with the binding site open to the luminal side of the Golgi membrane and sealed shut on the cytoplasmic side through the packing together of TM6 & TM7 against TM8 and TM9. Clear m$F_o$-D$F_c$ difference electron density was observed for the GMP ligand in three of the eight transporters in the unit cell (Supplementary Fig. 1). Unexpectedly the three GMP ligands adopt slightly different positions within the three binding sites (Fig. 1b), despite a r.m.s.d of only 0.45 Å between the three structures. In two of the three positions the GMP ligand adopts an extended conformation (Fig. 1c, d). However, in the third position captured, the GMP adopts a bent configuration, with the phosphate group pointing down towards the Golgi lumen (Fig. 1e). In chain C, the phosphate group of the GMP molecule forms a salt bridge with a conserved lysine (K289) on TM9 and interacts with a conserved tyrosine on TM1 (Y28) through a hydrogen bond (Fig. 1c). Lysine 289 forms part of the previously identified GALNK[289] motif, which plays an important role in discriminating between sugar groups of different nucleotide sugar molecules[10]. The interaction with Y28 is consistent with the essential role this side chain has in transport[10]. Moving along the GMP molecule, the ribose group also interacts with TM9 via another conserved tyrosine (Y281), which, although not strictly conserved, must be a bulky side chain to support transport[10]. Interestingly, the orientation of the ribose ring is significantly different to that observed in the GDP-mannose structure, resulting in an interaction between Y281 and the ring oxygen instead of the two hydroxyl oxygens (Supplementary Fig. 2). At the far end of the ligand the guanine ring is flipped 180° relative to that of GDP-mannose, but still sits within the previously observed nucleotide binding pocket. This pocket is characterised by the presence of a conserved FYNN[221] motif on TM7, which is required for guanine recognition. Similar conserved motifs are observed in other families of nucleotide sugar transporters in this region of the binding site, consistent with a general importance of TM7 in nucleotide recognition[11,13]. However, unlike the GDP-mannose ligand, GMP does not make a direct interaction to the conserved asparagines of the FYNN

motif, which are ~4.5 Å away. Instead the guanine interacts with a conserved serine (S266) on TM8, via an interaction with the ring nitrogen.

In chain D the GMP ligand adopts a similar extended position to that observed in chain C, with the guanine ring again flipped 180° relative to that observed in the GDP-mannose structure (Fig. 1d and Supplementary Fig. 2). Overall the interactions between the GMP in chain D and those in chain C are similar despite the different orientation of the ribose ring (Fig. 1b). S266 interacts with the ring nitrogen on the guanine base, Y281 with the ribose ring oxygen, while K289 interacts with the phosphate group. However, a surprising finding from our analysis of the GMP co-crystal structures was the unusual position of the GMP ligand in chain E (Fig. 1b). Unlike the position of the GMP ligand in chains C and D, the molecule adopts a bent configuration, with the phosphate group now orientated towards the luminal entrance of the binding site (Fig. 1e). In this position the phosphate now interacts with a conserved serine on TM1 (S32), rather than K289, which adopts a different rotamer configuration compared to chains C and D (Supplementary Fig. 3). The ribose group sits in a similar location within the binding site to that observed in chains C and D, but now Y281 interacts with the O2 hydroxyl rather than the ring oxygen, similar to that observed previously with GDP-mannose (Supplementary Fig. 2). In this position the guanine ring now sits further into the nucleotide binding pocket compared with the carbonyl group positioned near to asparagine N221 of the FYNN[221] motif. An additional interaction is also observed between the ring nitrogen of the nucleotide (N3) and a conserved serine on TM8 (S269), which was previously observed interacting with the ribose hydroxyl oxygen in the GDP-mannose structure. Taken together, the different positions of GMP within the crystal structure suggested that Vrg4 recognises GMP differently to GDP-mannose, which as we discuss below, is important for understanding how these transporters discriminate between nucleoside monophosphate and nucleotide sugar in the cell.

**Specific residues discriminate between ligands**. Although GMP adopts three different positions within the binding site, it is notable that in all three poses an interaction with Y281 on TM9 is observed. Previously we showed that a conservative mutation to phenylalanine retained activity whereas an alanine variant was non-functional[10]. However, given the prominent role that Y281 plays in GMP recognition we performed a more in-depth $IC_{50}$ analysis on the Y281F variant (Fig. 2a). We observed that the affinity of this variant for GDP-mannose remains the same as wild type (WT; 7.6 μM[10]), however the $IC_{50}$ for GMP increased to 23 μM, indicating a hydroxyl at this position aids recognition of the GMP ligand. Given that Y281 appears to discriminate between the two ligands, we sought to identify additional side chains that are important for GMP transport. Tyrosine 28 is essential, and its replacement with alanine abolishes transport, and K289 is important for sugar recognition and transport[10]. However, our structures also identified serines 266 and 269 as interacting with the guanine base. Interestingly, we observed a similar result to that of Y281 with S266, with an alanine variant resulting in a significant increase in the $IC_{50}$ value for GMP transport, whilst the affinity for GDP-mannose was only slightly reduced (Fig. 2b). The similar affinity for GDP-mannose as WT in this mutant is most likely due to the additional interactions formed between the GALNK motif and the mannose moiety compensating for the loss of interaction sites on the guanine base. Removal of S269 on the other hand did not result in any significant change in the $IC_{50}$ for either GMP or GDP-mannose (Fig. 2c). Our analysis of the binding site shows that Vrg4 is able

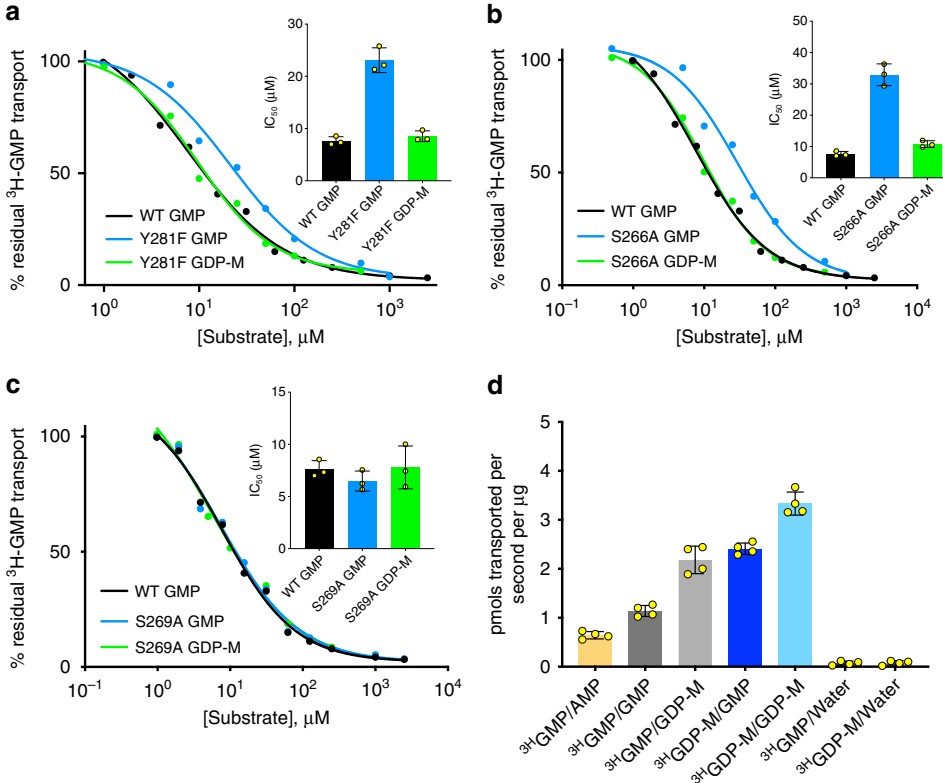

**Fig. 2** Characterisation of the GMP binding site. **a**, **b**, **c** Representative IC₅₀ curves determined for both GMP and GDP-mannose in the Y281F, S266A, and S269A variants, respectively. Insets show the calculated IC₅₀ value from three independent experiments, errors shown are standard deviations (s.d) of the mean. **d** Transport assays showing the highest level of transport occurs when GDP-mannose is present on both sides of the liposome membrane. $n = 4$ independent experiments, error bars s.d. Source data are provided as a Source Data file

to discriminate between GMP and GDP-mannose ligands and that different sets of residues are important for transport of either ligand.

**Structural explanation for transport rate differences.** NSTs are obligate antiporters, requiring the movement of one ligand in exchange for another. In the cell NSTs function to shuttle nucleotide sugars into the lumen of the ER or Golgi, in exchange for the cognate nucleoside monophosphate[7]. An important question has been how these systems ensure the direction of transport for their ligands. Previously we showed that Vrg4 functions at different rates depending on the ligands it is moving across the membrane, with GDP-mannose:GMP being more efficient than GMP:GMP in liposome-based assays[10]. The GMP co-crystal structures now provides an explanation for this phenomenon. It appears that GMP is simply less efficient at docking into the binding site and triggering transport compared to the larger GDP-mannose, which interacts with more sites in the protein. However, if this structural hypothesis is correct and the observed rate difference is caused by a more flexible binding position, we would predict that GDP-mannose:GDP-mannose antiport via Vrg4 would have a faster transport rate due to the lack of alternative interaction positions available with this larger ligand. Similarly, a ligand that makes fewer interactions than GMP, such as AMP (discussed below), would be expected to have a slower rate. We tested this hypothesis using a liposome-based assay and monitoring transport of both radiolabelled GMP and GDP-mannose. We observed the fastest transport rate when GDP-mannose is transported against itself and the slowest for GMP:AMP antiport (Fig. 2d and Supplementary Fig. 4),

confirming that the transport rate of Vrg4 correlates with the number of interactions made between the transporter and substrate.

**Structural basis of nucleotide specificity.** The features that underpin substrate selectivity in the SLC35 family are unclear, with sequence identity being a poor predictor of ligand recognition[9]. However, previously identified putative sequence motifs in Vrg4 appear to correlate with substrate specificity, which may facilitate sequence based functional assignment within the SLC35 family[11,13]. Specifically, in Vrg4 the FYNN[221] motif located on TM7 was linked to recognition of the guanine base. Vrg4 shows strict substrate specificity with respect to the nucleotide moiety, being able to recognise only purine bases (Supplementary Fig. 5). Within the purine bases, the native substrate GMP is transported with a much higher affinity (IC₅₀ 7 μM) compared to AMP (IC₅₀ 50 μM). It is known that the asparagines in the FYNN[221] motif are important for ligand recognition, but it is unclear what role they play[10]. To further understand this role, we analysed their ability to discriminate between the different purine bases. Alanine variants of both N220 and N221 resulted in a higher IC₅₀ for GMP than the WT protein; N220A 12 μM and N221A 22 μM (Fig. 3a). This result suggests that both asparagines are used to recognise and position the amino and carbonyl groups on the GMP ligand, and explains why AMP, with only one amino group is recognised at a much lower affinity than the WT protein. Analysis of the IC₅₀ of the mutants for AMP however, shows that for the N220A variant the affinity for AMP is markedly reduced, suggesting that the remaining asparagine, N221 cannot interact as well with the amine of AMP as it can with the carbonyl of GMP.

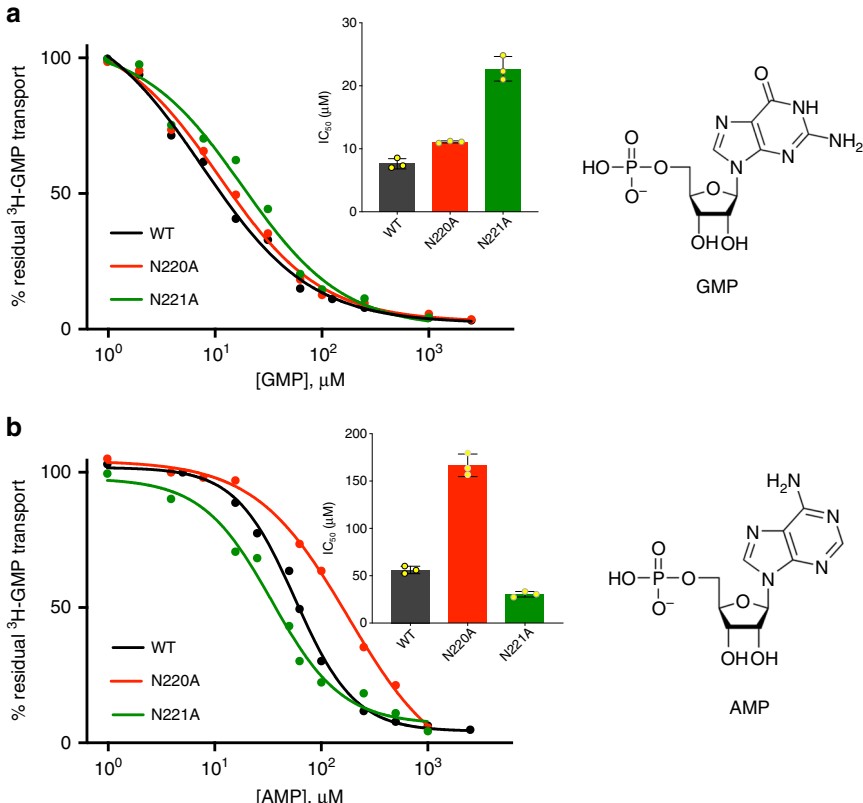

**Fig. 3** The FYNN motif mediates nucleotide discrimination. **a** Representative $IC_{50}$ curves determined for GMP in wild type (WT), N220A and N221A variants. **b** Representative $IC_{50}$ curves determined for AMP in wild type (WT), N220A and N221A variants. The chemical structures of GMP and AMP are shown for reference. Insets show the calculated $IC_{50}$ value from three independent experiments, errors shown are s.d. of the mean. Source data are provided as a Source Data file

However, the N221A variant has $IC_{50}$ values for AMP and GMP which are very similar to each other, 27 μM and 22 μM respectively, thereby no longer discriminating between these nucleotides (Fig. 3b). This result thus demonstrates that within the FYNN[221] motif, N221 plays a more significant role in purine selectivity. These biochemical data are supported by the structural comparison between the GMP and the GDP-mannose structures (Supplementary Fig. 2), which show that a similar feature observed in all conformations captured is the position of the carbonyl group close to N221.

**Lipids play an important role in mediating dimerisation.** NSTs involved in the transport of UDP-GalNac, GDP-fucose, PAPS (3′ phosphoadenosine-5′-phosphosulphate) and GDP-mannose are reported to form homo-oligomers ranging from dimers to hexamers[17–19]. However, the significance of oligomerisation and the role of lipids in regulating oligomeric state within the NST family remain unknown. Analysis of the apo, GMP and GDP-mannose bound structures of Vrg4 show the presence of well−ordered monoolein lipid molecules at a potential dimer interface. The dimer interface is formed between TM5 and 10 and contains two well-ordered monoolein lipid molecules, which contribute ~60% to the total buried surface area of 1514 Å$^2$ (Fig. 4a)[20]. Although Vrg4 crystallises as a dimer in monoolein, in detergent the protein is monomeric[10], raising the question of whether lipids induce dimer formation in Vrg4 and what implication dimerisation may have for function. To test the impact of the lipid environment on stabilisation of the protein we used a thermal shift assay, which has been used to test membrane protein lipid interactions[21]. We noticed that in a lipid environment Vrg4 has a significantly

higher melting temperature (~55 °C) than with either the detergents decylmaltoside (DM) (33 °C) or dodecylmaltoside (DDM) (38 °C) (Supplementary Fig. 6a). This large increase in melting temperature in the presence of lipid could be indicative of further stabilisation due to oligomerisation. We also observe possible dimer formation in SDS PAGE analysis of Vrg4 reconstituted into liposomes, which is not seen when the protein is in detergent, even at high concentrations of protein (Fig. 4b and Supplementary Fig. 6b). This raised the question of whether Vrg4 is a functional dimer in the membrane. To investigate whether lipids can induce higher order oligomer formation we performed glutaraldehyde crosslinking using protein purified under lipid rich conditions. A dimer band can be observed in SDS–PAGE which increases in intensity in the presence of both crosslinking agent and additional yeast polar lipids, indicating that Vrg4 can form higher order oligomers in the presence of lipid (Supplementary Fig. 6c).

To further understand the role of lipids within the dimer interface of Vrg4 we used molecular dynamics to embed Vrg4 in a membrane composed of 1,2-dipalmitoyl-sn-glycero-3-phosphocholine (DPPC) and 1,2-dimyristoyl-sn-glycero-3-phosphocholine (DMPC). These simulations confirm that phospholipids accumulate within the dimer interface observed in the crystal structure (Fig. 4c, d). Interestingly, the MD simulations show that the dimer interface can accommodate four DPPC lipids, arranged as a bilayer (Supplementary Fig. 6d). This indicates that in the native membrane environment the Vrg4 dimer will be held together through stronger interactions, than in a detergent micelle, consistent with our cross-linking analysis.

To test the functional significance of dimerisation we developed a dominant negative biochemical assay (Fig. 4e and

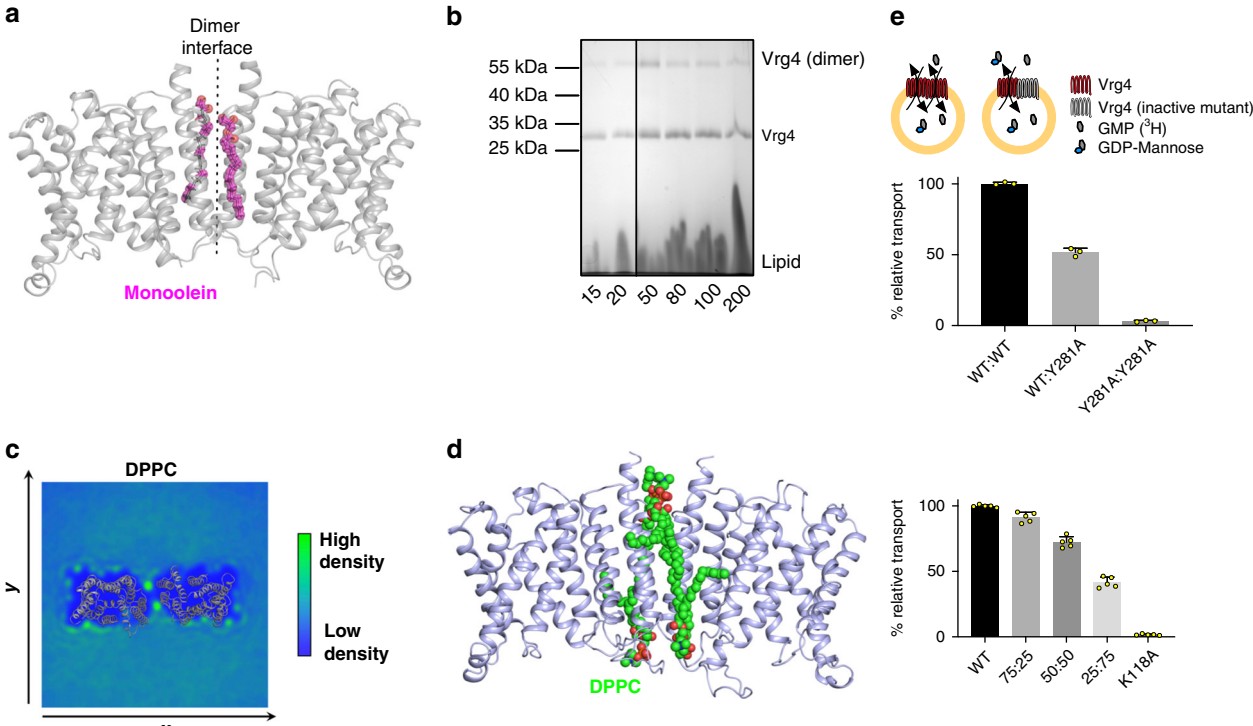

**Fig. 4** Lipid mediated dimerisation of Vrg4. **a** Vrg4 dimer observed in the crystal structure with two monoolein lipid molecules (magenta), packed within the dimer interface (dashed line). **b** SDS–PAGE analysis of Vrg4 reconstituted at different ratios of protein to lipid, showing the presence of a dimer band at ~ 55 kDa. **c** Density plot of DPPC lipids around the Vrg4 dimer, in a slice perpendicular to the membrane. Data were gathered using ca. 40 μs of CG MD simulation, and calculated with the VMD volmap plugin. Specific green regions highlight density present at the primary lipid binding region in the dimer interface. The Vrg4 dimer is shown in cartoon and overlaid for reference **d** Vrg4 dimer observed in the simulation shown in **c** with four DPPC lipids (green) within the dimer interface shown. **e** Dominant-negative assay showing the functional unit of Vrg4 is monomeric, and that Vrg4 forms dimers in a liposome. $n = 3$ (upper panel) or 5 (lower panel) independent experiments, errors shown are s.d. Source data are provided as a Source Data file

Supplementary Fig. 6e). We reasoned that if equimolar amounts of functional and non-functional Vrg4 were reconstituted into liposomes, then random mixed dimers would form, containing both active and inactivate transporters. If Vrg4 is a functional dimer we would expect to see a greater than 50% reduction in transport. However, we observed that mixing fully functional WT protein with an inactive mutant, Y281A, discussed above, resulted in exactly 50% transport activity. This result implies that each monomer of Vrg4 acts as an independent functional unit, similar to other dimers of SLC transporters[22,23]. We then repeated this experiment, but instead used a K118A variant, which is transport inactive but binding competent (Supplementary Fig. 6f), and observed ~75% activity (Fig. 4e). Reconstituting at different ratios (75 and 25% WT) also gave a higher than expected transport rate, confirming that Vrg4 forms oligomers within the liposome membrane (Fig. 4e). Our interpretation of this data is that when Vrg4 is in a mixed dimer, where one subunit is unable to transport and in a locked conformation, the WT subunit is able to cycle faster. This effect is most likely due to the increased stability being imparted by an immobile partner, which provides a stable platform against which the active transporter can move. This phenomenon was also recently shown for the SLC26 anion transporters, which despite being a different family also form structural oligomers in the membrane[24]. In this study a similar effect was observed, where the mixing of active and inactive monomers created dimers with great than 50% activity. Together with the structural, cross linking and MD data, we propose that Vrg4 dimerises through an interface mediated by lipid molecules, which form an integral part of the dimer.

**Structural basis for short chain lipid dependence.** The Golgi membrane is known to have a different lipid composition in comparison to the plasma membrane, which gives rise to a thinner bilayer thickness[25]. Vrg4 transport activity is dependent on short chain lipids to function, with a severe drop in activity observed in longer chain lipids, such as 1-palmitoyl-2-oleoyl (PO) lipids, which consist of one 16 and one 18 carbon fatty acid chain[10]. However, the structural basis for this observation remains obscure. This phenomenon could be due to hydrophobic matching, especially given the short length of the hydrophobic region of Vrg4. Indeed, when Vrg4 is inserted into a coarse grained DPPC membrane, it results in the thinning of the bilayer to more resemble the thickness of a DMPC bilayer (Supplementary Fig. 7). Our simulation data from the mixed DMPC/DPPC bilayers also showed the accumulation of lipid at an additional site on the transporter (Fig. 5a, b). This second lipid binding site only accommodates DMPC and not DPPC, and occurs in a shallow groove between TMs 1, 9, and 10 (Fig. 5c and Supplementary Fig. 8). The preference for DMPC over DPPC at this site suggests lipids of 16 carbon chain lengths and above are excluded, whereas the shorter DMPC lipid, with only a 14 carbon acyl chain, can be accommodated. Based on a repeat swapped model of the cytoplasmic facing state of Vrg4, we previously suggested that TM9 is likely to play an important role in the alternating access mechanism[10]. TM9 was also identified as being important for the transition between luminal and cytoplasmic facing conformations of the plant triose phosphate antiporter, TpT, a distant homologue of the NST family[26,27]. These results suggest that short chain lipids may allow TM9 to undergo the

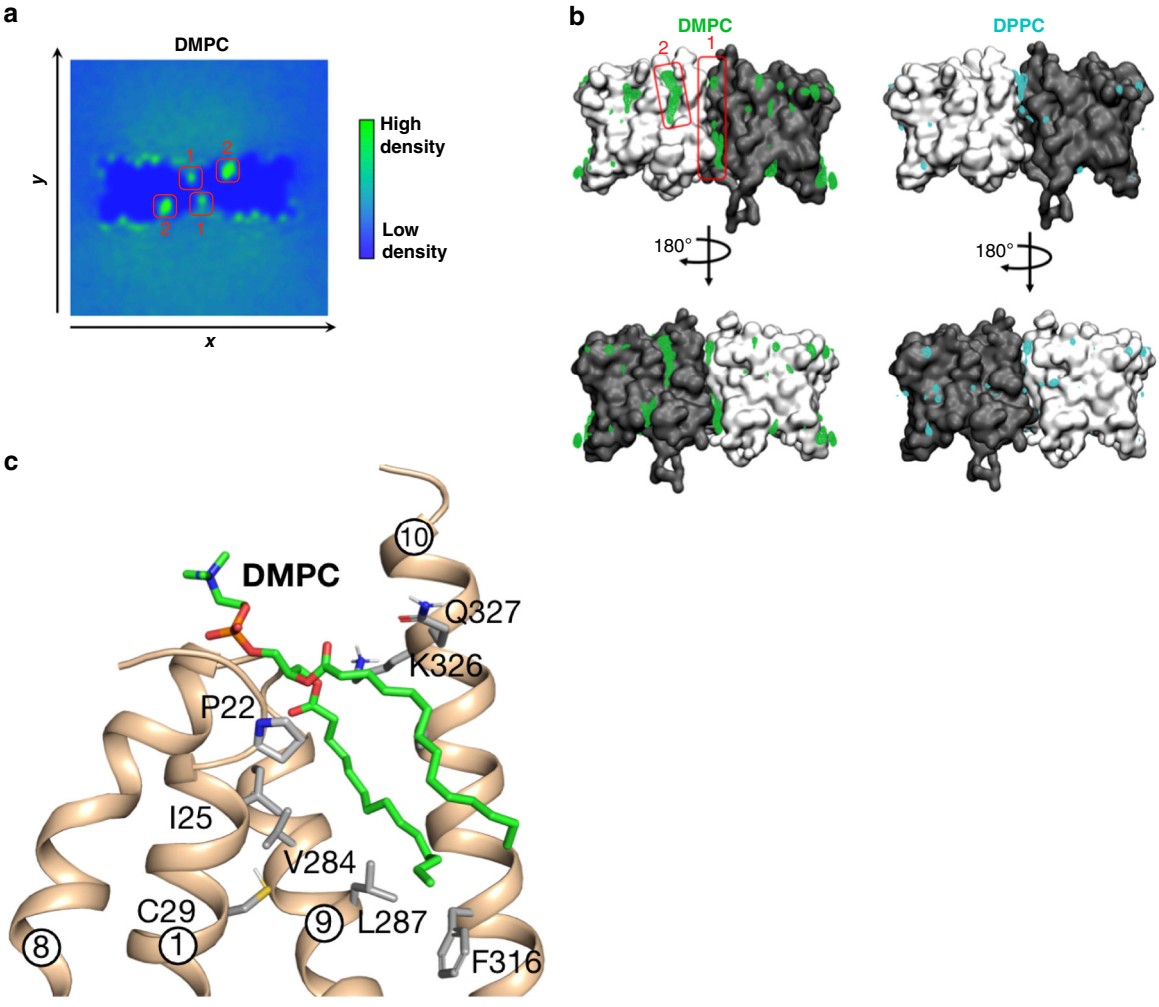

**Fig. 5** Identification of short chain lipid binding site. **a** Density plot of DMPC lipids around the Vrg4 dimer. In addition to the primary lipid binding sites on the dimer interface (labelled '1'), secondary binding sites can be observed (labelled '2') (green densities). **b** Views of Vrg4, with the densities from **4c** and **5a** shown as coloured mesh. The protein is shown as surface, with each Vrg4 monomer coloured as white or grey. The primary binding site – marked '1' in the top left view – is occupied by both DMPC and DPPC. A secondary site – marked '2' – is exclusively occupied by DMPC. **c** View of the bound DMPC lipid in the secondary binding site, following 30 ns of atomistic simulation. Residues interacting with the lipid are indicated as sticks and labelled and helices are labelled

conformational changes necessary to cycle between inward and outward facing states during transport.

## Discussion

Nucleotide sugar transporters function as obligate exchangers, swapping the activated sugar molecule in the cytoplasm for the spent nucleoside monophosphate in the ER and Golgi lumen. NSTs consist of 10 TM helices that are arranged as two inverted topology repeats of five TMs, such that TM1-5 can be superimposed on TM6-10 via a 180° rotation in the plane of the membrane[10]. Previously we identified that transport within the NSTs occurs through the switching conformation of the first four helices that form each of the five TM bundles. Two pairs of helices on either side of the binding site function as mobile gates, which rock around the central binding site, dictating whether the binding site is accessible to either the cytoplasm (inward facing) and ER/Golgi lumen (outward facing). A long-standing question in the NST field has been whether a mechanism exists whereby NSTs facilitate the direction of transport, such that adequate supplies of nucleotide sugars are available to the glycosyltransferases. The recent discovery that Vrg4 indeed does exhibit different rates of transport, such that GDP-mannose is

transported across the membrane with a faster rate than GMP, demonstrates the internal mechanics of transport could play a supporting role to the established concentration gradients across these membranes. Several lines of evidence, including crystal structures and steered MD of a related plant triosephosphate-phosphate antiporter, TpT, along with our own analysis of a cytoplasmic facing model of Vrg4[10], suggest that conserved side chains on the main gating helices are likely to facilitate reorientation of the transporter from inward to outward facing states and vice versa following interactions with the bound ligands[13,26]. Our findings that GMP adopts several similar, yet discrete conformations in the binding site now provides evidence that the slower rate of transport observed for GMP vs. GDP-mannose is due to less effective coupling between the ligand and the transporter. The complex with GMP shows the guanine ring interacts far less with the conserved FYNN[221] motif of TM7, which is responsible for distinguishing between adenine and guanine (Fig. 3). In addition, the ribose group also appears to adopt several conformations in the GMP structures, with only one forming a strong interaction with Y281 on TM9, which is essential for transport and one of the key gating helices[13]. Interestingly, we noticed that K289 also on TM9, adopts a

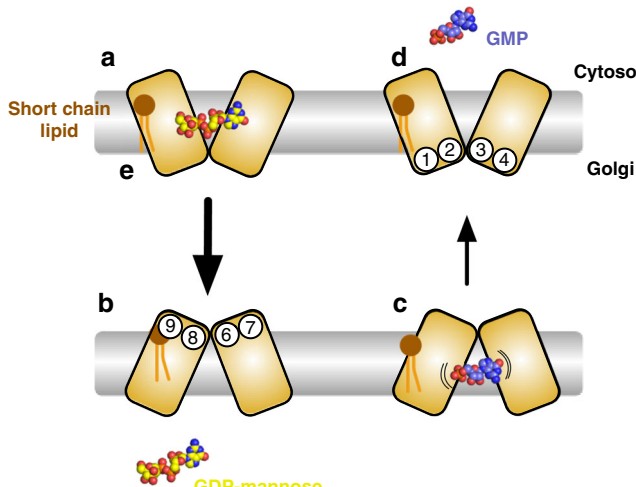

**Fig. 6** Model of alternating access mechanism in Vrg4. **a** Vrg4 adopts an inward open state with the binding site orientated towards the cytosol. GDP-mannose binds, resulting in the fast interconversion to the outward open state **b**, facing the Golgi lumen. **c** Following release of the GDP-mannose, GMP binds the transporter. The smaller size of GMP results in less efficient transport, resulting in a slower interconversion back to the inward open state **d** The key gating helices are TM1, 2 and 3, 4 that alternate around the central binding site. **e** Short chain lipids are necessary to facilitate the movement of the gating helices, with the transporter being inactive in longer chain lipids

substantially different rotamer configuration between the GMP and GDP-mannose structures (Supplementary Figs. 2 and 3). In the GDP-mannose structure K289 makes an interaction with S32 on TM1 and the beta phosphate, functioning to connect these two gating helices with GDP-mannose recognition. However, in the GMP structure we observe K289 extending away from TM1, and out towards the alpha phosphate group of GMP. Only two of the three GMP molecules interact with K289, with the third GMP adopting a bent configuration that orientates the alpha phosphate towards the luminal gate (Fig. 1e). The inability of GMP to engage with K289 efficiently may explain why transport occurs at a reduced rate. Given the important role that TM9 plays in transport, it was intriguing to discover a lipid binding site connecting this helix to TM1. This would provide a further structural link between the pairs of gating helices within the transporter, providing an explanation for the short chain lipid dependence. Taken together, these insights suggest a model for transport within the NSTs, with an important role for the ligand in facilitating the rates of transport across the ER and Golgi membranes (Fig. 6). The larger GDP-mannose ligand is able to dock more efficiently in the binding site, resulting in faster transport, whilst the smaller GMP is more mobile, requiring additional time to efficiently interact with the pivot point residues and initiate transport.

Finally, lipids are emerging as important regulators of secondary active transporters[28,29]. Our recent discovery that short chain lipids have a role in regulating Vrg4 activity demonstrates a specific role of lipids in this family of transporters and within the dynamic membranes of the secretory pathway in eukaryotes. Following the insertion of Vrg4 into lipid bilayers we observed the presence of four well-ordered lipid molecules at the dimer interface of Vrg4 (Fig. 4d). Within the SLC35 family, oligomerisation has been reported for several members, and evidence from cell biology studies show that heterooligomers can be formed between different NST monomers and N-acetylglucosaminyl-transferases, potentially as a route to regulation[17,19,30].

Interestingly, in Vrg4 we observe a dynamic oligomerisation interface, where lipids appear to control dimer formation. This opens up the exciting prospect that changing lipid environments within the secretory pathway may function to regulate NST oligomerisation and activity, inserting an additional level of control on cellular glycosylation within the cell.

## Methods

**Protein expression and purification**. The gene encoding ScVrg4 (Uniprot P40107) was amplified from *Saccharomyces cerevisiae* genomic DNA and cloned into the pDDGFP-Leu2D vector (addgene 102334). Standard site directed mutagenesis techniques were used to make variant forms of Vrg4. Wild type and variant proteins were produced in *S. cerevisiae* strain BJ5460 (ATCC 208285) and purified using standard nickel affinity chromatography. Membranes were thawed and solubilised in purification buffer which consisted of, 1 × PBS containing an additional 150 mM NaCl and 10% glycerol and 1% n-dodecyl−β-D-maltopyranoside (DDM, Glycon) with stirring for 1.5 h. The solubilised material was recovered through ultracentrifugation at > 200,000 × *g* for 1 h. A final concentration of 18 mM imidazole was added and the protein was bound to nickel resin (GE Healthcare) in batch for 4 h. The resin was washed with purification buffer containing first 18 mM imidazole and then 25 mM imidazole and 0.2% DDM for 15 and 25 column volumes respectively. Vrg4 was eluted from the resin with purification buffer containing 250 mM imidazole. TEV protease was added and the protein was dialysed overnight in gel filtration buffer containing 0.03 % DDM (20 mM Tris pH 7.5, 150 mM NaCl). After dialysis, the protein was passed through a HisTrap column to remove the TEV protease and the GFP tag. The pure protein was concentrated using a vivaspin 50,000 MWCO spin concentrator. Protein for crystallisation was applied to a Superdex 200 10/300 gel filtration column equilibrated in a buffer consisting of 20 mM Tris–HCl pH 7.5 and 150 mM NaCl with 0.03% DDM, for reconstitution the detergent was changed to 0.3% n-decyl-β-D-maltopyranoside.

**Protein purification and glutaraldehyde crosslinking**. For the cross-linking experiments Vrg4 was purified from membranes in purification buffer, consisting of 1 × PBS, 150 mM NaCl, 10% glycerol and 1% n-dodecyl−β-D-maltopyranoside (DDM, Glycon) whilst stirring for 1.5 h at 4 °C. The solubilised material was recovered through ultracentrifugation at > 200,000 × *g* for 1 h. A final concentration of 18 mM imidazole was added and the protein was bound to nickel resin (GE Healthcare) in batch for 4 h. The resin was washed with purification buffer containing 18 mM imidazole and followed by a second wash with 25 mM imidazole containing 0.1% DDM for 8 and 10 column volumes respectively. Vrg4 was eluted from the resin with purification buffer containing 250 mM imidazole. TEV protease was added and the protein dialysed overnight in gel filtration buffer containing 0.015% DDM (20 mM Tris pH 7.5, 150 mM NaCl). After dialysis, the protein was passed through a HisTrap column to remove the TEV protease and the GFP tag. The pure protein was concentrated using a vivaspin 50,000 MWCO spin concentrator to 0.5 ml and applied to a Superdex 200 10/300 gel filtration column equilibrated in a buffer consisting of PBS with 0.15 % DM.

For crosslinking 6 μg of protein were incubated in PBS with either 10 or 20 μg yeast polar lipids (also in PBS and extruded through a 0.4 μm filter) for 30 min at 20 °C in a 10 μl volume. A final concentration of 0.2% glutaraldehyde was added and the reaction left for a further 20 min prior to the addition of 1 μl 1 M tris to quench the reaction. Samples were loaded onto a 12% SDS–PAGE gel and stained with Coomassie blue.

**Crystallisation**. Crystallisation was performed using protein at 40 mg ml⁻¹ final concentration, as determined using absorbance at 280 nm. In total 10 mM GMP was incubated with the protein on ice for at least 2 h prior to LCP set up. Protein-laden mesophase was obtained by mixing monoolein with protein in a 60:40 (w/w) ratio using a coupled syringe device (Art Robbins, USA). Crystals appeared at 4 °C in an optimised crystallisation screen consisting of 26–30% (v/v) PEG 400, 0.1 M sodium citrate pH 5.0 and 75 mM sodium chloride or sodium acetate. Crystals grew and within 4 days after initial set up the top glass plate was removed with a glass scribe and 2 μl of crystallisation solution containing 20 mM GMP was added. The well was resealed using a thin glass coverslip and left for at least 18 h. Crystals were harvested using 30 μm micromounts before flash cooling in liquid nitrogen.

**Structure determination**. X-ray diffraction data were collected at beamline Proxima 2a at Soleil, France to 3.39 Å resolution. Indexing and integration were performed with XDS[31], followed by scaling and merging with AIMLESS[32]. Initial phases were obtained by molecular replacement (MR) using PHASER[33] in the CCP4 suite[34]. The search model was the previously determined crystal structure of Vrg4 (PDB: 5OGE). Model building into the electron density map was performed with COOT[35], with structure refinement carried out in PHENIX[36]. Model validation was carried out using Molprobity[37]. Images were prepared using PyMol.

**Protein reconstitution into liposomes**. Vrg4 was reconstituted into liposomes made from yeast polar lipids using the dilution method into preformed lipid vesicles[38]. Chloroform was removed from yeast polar lipids (Avanti polar lipids) through the use of a rotary evaporator to obtain a thin film. The lipids were washed twice in pentane and then resuspended at 5 mg ml⁻¹ in lipid buffer (20 mM HEPES pH 7.5, 100 mM KCl). These lipid vesicles were frozen and thawed twice in liquid nitrogen and stored at −80 °C until required. For reconstitution, the lipids were thawed and then extruded first through a 0.8-μm filter and then through a 0.4-μm filter. Purified Vrg4 in DM (at 0.5 μg μl⁻¹ concentration) was added to the lipids at a final lipid:protein ratio of 80:1 and incubated for 1 h at room temperature, then for a further 1 h on ice; for the no-protein liposome control, the same volume of gel filtration buffer containing 0.3% DM was added. After this time, the protein–lipid mix was diluted rapidly into 65 ml of assay buffer (20 mM HEPES pH 7.5, 50 mM KCl, 2 mM MgSO₄) and proteoliposomes were collected through centrifugation at > 200,000 × g for 2 h. To remove trace detergent the proteoliposomes were dialysed overnight against a large volume of assay buffer. After dialysis, the proteoliposomes were collected and resuspended in assay buffer to a final protein concentration of 0.25 μg μl⁻¹, and then subjected to three rounds of freeze–thawing in liquid nitrogen before storage at −80 °C. The amount of protein (both wild-type and mutant variants) reconstituted into the lipids was quantified by SDS–PAGE and densitometry.

For the dominant negative assays the required ratios of wild type and mutant variants were mixed in detergent, left for one hour on ice prior to reconstitution as above. In order to validate the data from these experiments the reconstitution of the different ratios were repeated in duplicate from separate purifications of both wild type and mutant proteins.

**Transport assays**. To analyse transport activity, proteoliposomes were thawed and the desired concentration of internal substrate was added (typically 0.5 mM for mutant variant analysis). To load the liposomes, they were subjected to six rounds of freeze–thaw in liquid nitrogen and then extruded through a 0.4-μm membrane. The equivalent of 1 μg of protein was added to 50 μl of assay buffer containing 0.5 μM ³H-GMP, which initiated the transport. All assays were performed at 30 °C unless stated otherwise. The uptake of radiolabelled substrate was stopped at the desired time by rapidly filtering onto 0.22-μm filters, which were then washed with 2 × 2 ml cold water. The amount of GMP transported inside the liposomes was calculated by scintillation counting in Ultima Gold (Perkin Elmer)

For IC₅₀ calculations the liposomes contained 0.5 mM GMP on the inside and the external buffer contained the desired amount of cold competitor. The uptake of radiolabelled substrate was stopped at the desired time (3 min) by rapidly filtering onto 0.22-μm filters, which were then washed with 2 × 2 ml cold water. The amount of GMP transported inside the liposomes was calculated by scintillation counting in Ultima Gold (Perkin Elmer). For IC₅₀ values the whole experiment was repeated in triplicate to calculate the mean and s.d. For every experiment a control using WT protein was conducted alongside, however the data shown in the figures was data published previously[10]. A representative curve is shown for each main figure, with the mean and s.d. shown in a bar chart. All the IC₅₀ curves are shown in supplementary data (Supplementary Fig. 9 and 10).

For the analysis of rates under different antiport conditions, liposomes were loaded with 0.5 mM of the substrate (or 1 mM for AMP) on the inside and 20 μM of either GDP or GDP-mannose, with a trace amount of 3 H substrate on the outside. Time points were taken to produce an initial linear rate (Supplementary Fig. 4). The amount of substrate transported at 45 s is shown in the main figure. These experiments were performed at 22 °C.

For the mixed dimer analysis, a total of 1 μg of protein was added to 50 μl of assay buffer containing 0.5 μM ³H-GMP, which initiated the transport. Transport was monitored over time with time points taken at 10, 20, 30, and 60 min. The main figure shows the relative transport at 45 s. Each experiment was repeated in triplicate. In addition, the entire analysis was repeated using protein from separate purifications.

**Thermal stability measurements**. Thermal stability of protein samples was analysed using a Prometheus NT.48 (NanoTemper Technologies)[39]. The proteins were diluted to a final concentration of 0.7 mg ml⁻¹ into buffer containing 20 mM Hepes, pH 7.5, 50 mM KCl, 2 mM magnesium sulphate and 0.03% (w/v) DDM or 0.3% DM. Thermal measurements were carried out in a range from 20 to 95 °C with 1 °C per min steps. The resulting melting curves were generated by plotting the first derivative of the fluorescence ratio at 330 nm/350 nm (excitation 280 nm) against temperature. For stability in the presence of ligand (GMP) the ligand was added to the sample at a final concentration of 0.5 mM and the sample was incubated at room temperature for 5 min prior to analysis. For the sample in liposomes, reconstituted Vrg4, at a protein concentration of 0.4 mg ml⁻¹, in yeast polar lipids were prepared in 20 mM Hepes, pH 7.5, 50 mM KCl, 2 mM magnesium sulphate and extruded through a 0.4 μm membrane prior to analysis. For the solubilised liposome sample, liposomes were solubilised in 1.5% DDM for one hour and centrifuged at 100,000 × g for 30 min prior to analysis.

**Coarse grained molecular dynamics**. Vrg4 dimer coordinates were converted to the Martini 2.2 protein representation[40]. In addition to the bonds implicit in the

Martini force field, elastic bonds of 1000 kJ mol⁻¹ nm⁻² were applied between protein backbone beads within 1 nm across the full Vrg4 dimer. The protein was then built into a lipid bilayer composed of 875 dipalmitoylphosphatidylcholine (C16:0; DPPC) molecules and 875 dimyristoylphosphatidylcholine (C14:0; DMPC) molecules, distributed equally between the two leaflets. Note that, due to the coarse nature of the Martini force field, the DPPC and DMPC lipids also represent C18:0 and C12:0 tails respectively. The membranes were built using the insane method[41] and solvated with Martini water and Na and Cl to 150 mM. Three independent repeats, each from an independently constructed membrane, were run out to 12–15 μs each, using 20 fs time steps, in the NPT ensemble at 323 K with the V-rescale thermostat and semi-isotropic Parrinello–Rahman pressure coupling[42]. For analysis purposes, the first 4 μs were discarded. Densities for both the DPPC and DMPC lipids were computed using the volpam utility in VMD[43] and Gaussian smoothed with a resolution of 0.2 nm. Densities were averaged over the three repeats for the main figures.

For the bilayer thickness simulations, Vrg4 dimers were built into membranes comprising just DPPC or DMPC lipids. Bilayer thicknesses were assessed over the duration of > 3 μs CG simulations.

**Atomistic simulations**. A post-15 μs snapshot of the DPPC/DMPC CG data was trimmed in size and converted to an atomistic description[44], using the CHARMM36 force field[45] modified to include virtual-sites[46]. Missing loops were added using SWISS-MODEL[47]. The systems were solvated with TIP3P water and Na⁺ and Cl⁻ ions were added to 150 mM. The systems were energy minimised using the steepest descents method, then equilibrated with positional restraints on heavy atoms for 100 ps in the NPT ensemble at 310 K with the V-rescale thermostat and semi-isotropic Parrinello–Rahman pressure coupling[42]. Production simulations were run without positional restraints, with 4 fs time steps over 30 ns. Lipid contact was analysed using MDAnalysis[48], where residue-lipid distances below 0.4 nm were considered as contacts.

All simulations were run using GROMACS 2018[49].

## Data availability

Data supporting the findings of this manuscript are available from the corresponding authors upon reasonable request. A reporting summary for this Article is available as a Supplementary Information file. Atomic coordinates for the crystal structure have been deposited in the Protein Data Bank under accession number 6QSK. The source data underlying Figs. 2a-d, 3, 4e and Supplementary Figs 4, 5, 9 and 10 are provided as a Source Data file.

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

## Acknowledgements

We thank the staff of beamlines Proxima-2A, Soleil, France for assistance in data collection. This work was supported by Wellcome (102890/Z/13/Z, 102890/Z/13/A) and the Medical Research Council, UK (MR/S021043/1) awards to SN and J.L.P, and Wellcome (208361/Z/17/Z), Medical Research Council (MR/S009213/1) & Bio-technology and Biological Sciences Research Council, UK awards (BB/P01948X/1, BB/R002517/1 BB/S003339/1) to P.J.S. ARCHER UK National Supercomputing Service (http://www.archer.ac.uk), provided by HECBioSim, the UK High End Computing Consortium for Biomolecular Simulation (hecbiosim.ac.uk), which is supported by the EPSRC (EP/L000253/1).

## Author contributions

J.L.P. & S.N. conceived the study. J.L.P., R.A.C., P.J.S. & S.N. designed the experiments and interpreted the data. J.L.P. and S.N. wrote the paper with assistance from R.A.C. & P.J.S.

## Competing interests

The authors declare no competing interests.
