## [Peer Review File · Nature Communications]

Reviewers' Comments:

Reviewer #1:

Remarks to the Author:

Parker et al present a structure of the nucleotide sugar transporter Vrg4 in the presence of GMP, which they compare with a previously determined structure in the presence of GDP-mannose. They use the observed structural differences in the ligand binding modes to interpret differences in kinetics of transport of the two substrates. In addition, they analyze the oligomeric state of the protein (likely dimeric in lipid bilayers) and reevaluate previous MD simulations to propose a role for lipids in regulation of transport. The work advances our knowledge in the transport mechanism.

The work is technically well done, but I have some doubt about the authors' (over)interpretation of the data.

Specific points (in no particular order)

Page4: There are 8 transporters in the unit cell, but only three contain GMP. Apparently, exactly the same three transporter molecules in each unit cell hold GMP. This must mean that that there are structural differences between the transporters in the unit cell that force preferential binding to some, or disfavor binding to the others. This must be discussed, and explained. On a similar note: It is always Chain E that has the unusual binding pose of GMP. What is special/different about Chain E? A careful analysis of differences may help the prediction and simulation of subtle changes in the binding site, and the dynamics thereof. At least this should be carefully discussed.

Page 8 and discussion: It is unclear to me what exactly the authors are trying to say about the kinetics of transport. First, from their previous paper, I understand that GMP and M-GDP display similar IC50 (~Km) values (Fig 2 in the Nature paper) but different Vmax (?) (deduced from Figure 1e of the Nature paper). This would be consistent with lower binding affinity for GMP than for M-GDP. In vivo, in the lumen of the Golgi/ER M-GDP concentrations will be low, because M-GDP is converted by enzymes upon entry. Therefore, there will just be GMP left to drive the antiport. This has not much to do with kinetics, but just with concentrations. On the cytoplasmic side of the membrane, the higher concentration of M-GDP and very low concentration of GMP favors use of M-GDP as substrate. To what extent do the kinetic differences (=differences in apparent affinities) play a role too? A kcat/Km determination could shed light.

Related: Fig 2d of the current manuscript shows that the kinetic differences between GMP and M-GDP are not consistent. The Grey and blue bars (3HGMP/GDP-M vs 3HGDP-M/GMP) are identical, which is difficult to reconcile with significant kinetic differences. This must be explained.

Fig 4b: the SDS-PAGE interpretation seems cherry picking to me: On the one hand the authors claim that in detergent monomers are favored (MS and SEC-MALS), even in the presence of lipids, but in SDS (also a detergent, but now a denaturing one) they claim that dimers are observed. In my view SDS band in the presence of lipid could just be a big artefact of denatured protein (as has been observed many times in the past).

Page 12: the mixed dimer effects need more explanation. I am not convinced about the 50% vs 75% difference. The statistics is not explained (Notwithstanding the ticked box in the reporting summary). How many biological replicates were used? It could also be that the K118A mutant reconstitutes poorly, because it may not be stable in the absence of substrate, and that the ratio of reconstituted proteins is not the same as the ratio of purified proteins in detergent solution. In any case this result need much more discussion. The way it is presented now seems pure speculation and overinterpretation.

P15 "substantially different rotamer configuration" Without showing densities, I am not convinced given the moderate resolution.

Reviewer #2:

Remarks to the Author:

The manuscript of Parker et al. follows up on Parker & Newstead, 2017, that detailed the structure of yeast GDP-mannose transporter, Vrg4, in a conformation open towards the golgi lumen and without ligand and with GDP-mannose bound. The novel structure presented in this manuscript details the Vrg4 structure in the same conformation but in the presence of GMP. This is a relevant structure as Vrg4 exchanges cytoplasmic nucleotide sugars for nucleoside monophosphate from the golgi lumen. The authors show that GMP, surprisingly, has two major binding modes in the Vrg4 conformation open to the golgi lumen. In addition, the interaction of certain lipids with the transporter is characterized computationally and the functional relevance of a potential dimer is studied, leading to the suggestion that the monomers in the dimer functionally interact. Though the structure of GMP-bound Vrg4 is novel and important, the additional sections of the manuscript on the characterization of the substrate binding, the relevance of lipids, and the functional interactions in the dimer are less convincing. The mutants indicated in Fig2 may contribute to discriminating between GMP and GDP-mannose, but these are very small at most, and negligible compared to the K289A mutant published previously (Parker 2017). The explanation on differences in transport rates is interesting, but speculative and confusing, given the large emphasis on substrate binding while the apparent affinities for both substrates are not different. The sections on the relevance of lipids are exclusively computationally and would benefit from functional studies to support the claims. The characterization of the dimer overall is interesting and relevant, but would require additional supporting evidence that this dimer truly exists. The manuscript of Parker et al. details several relevant and interesting topics on nucleotide sugar transporters. I find the current version of the manuscript a bit premature. The manuscript would benefit from a more clear description of the experiments and a clearer interpretation of the data. In addition, more extensive or additional experiments (see below) are required to make the claims stronger.

Concerning the functional and mechanistic interpretations of substrate binding I have a few concerns:

- 1) The functional experiments are insufficiently detailed. For every figure, it should be made clear what substrate is in what compartment and at what concentration. The current poor descriptions leave unwanted room for different interpretations.
- 2) The IC50 values measured in Fig2 and Fig3 differ only 2-4 fold. This is generally not considered as significant and within the range of experimental error. The authors should detail why this small difference is taken as relevant. Error bars should be added to every datapoint in all IC50 curves.
- 3) Control the direction (P8) > what is the evidence that Vrg4 actually controls the direction? An alternative and more straightforward interpretation is that the enzymes in the golgi and the cytoplasm contribute to high concentrations of GMP in the Golgi and GDP-mannose in the cytoplasm. These opposite concentration gradients may already suffice to "control the direction".
- 4) On P8 the authors compare the transport rate between GMP and GDP-mannose to conclude that GDP-mannose is transported more efficiently. The efficiency of transport is best reflected in the Vmax, which requires that measurements are done at saturating substrate concentrations. Was this the case for the data presented in Fig2d?
- 5) P14, bottom. The direction of transport / different rates of transport > see above. All transport measurements are done in proteoliposomes. Given the reconstitution method, it appears likely that the transporter is reconstituted in equal amounts in two different orientations. That implies that potential differences in substrate affinities on the cytoplasmic and golgi-luminal side could not be measured (this is also not claimed by the authors). As information on the asymmetry of the substrate binding sites in combination with the physiological concentrations of GMP and GDP-mannose in either compartment is absolutely required to allow any prediction on "directionality", the discussion on the different rates is irrelevant. An increased transport rate of GDP-mannose alone may support increased export from the golgi-lumen as well. Revise to indicate the modest

role of the rate with respect to the action of catabolytic and anabolytic enzymes and potential asymmetry in the substrate affinities on cytoplasmic and luminal binding sites.

6) P8+P15 the more dynamic GMP is simply less efficient at docking into the binding site > what is the evidence that GMP is less efficient at docking? Though the structures indicate multiple binding modes, the IC50, which is an indirect indicator for affinity, is identical for GMP and GDP-mannose. How do the authors reach the conclusion that the "recognition of GMP is less efficient than the GDP-mannose"?

Concerning the study on the oligomeric state and lipid-dependence of Vrg4 I have the following concerns:

7) The evidence that the protein is oligomeric in the membrane is exclusively based on the available structures that show dimers (next to monomers) in an orientation that physiologically could make sense. However, the dimer may be an artefact from the high protein concentrations and the 2D plane resulting from the LCP. Additional independent proof that Vrg4 forms dimers in membranes is absolutely required. As the tentative structure of the dimer is known, I suggest to perform cysteine cross-linking experiments (oxidative or using cross-linkers) for this purpose.

8) P11, is the protein active in bilayers composed of only DPPC and DMPC that were used for the MD?

9) P12, the studies on the functional significance of a potential dimerization is interesting, but not sufficiently convincing. Membrane reconstitution always leads to some experimental variation, the amount of functionally reconstituted protein will vary. In the setup chosen by the authors, the difference between cooperative and independent functioning essentially depends on the activity measured for one ratio. This is too little. Additional intermediate ratio's should be tested to make this result reliable.

10) P12, bottom. Stabilizing effect > it is not clear what is meant. The reference refers to a transporter that appears to use a very different transport mechanism than Vrg4. Clarify.

11) P13, top. In the preceding paper (Parker 2017) the effect of DMPC appears to be attributed to hydrophobic mismatch. In the current manuscript this effect appears assigned to a short chain lipid dependence, which is much more specific than hydrophobic mismatch. What is the basis for this novel interpretation of the data?

12) The observation of a stable interaction between DMPC and the protein in MD simulation is interesting. However, the mechanistic interpretation is far too early. If the authors wish to substantiate this claim, mutants should be made to disrupt this interaction followed by functional characterization. Alternatively, this section should be left out.

Additional comments

P4, line2 kinetic difference > specify whether this is Km or Vmax or both

sfig1 place the proteins in the three panels in the identical orientation to demonstrate the different ligand orientations

P5, bottom what is the rmsd between the GMP and the GDP-mannose structures?

P6, bottom affinity of this variant for GDP-mannose remains the same as wildtype > Fig2a does not depict any data for wt. Is this value based on literature?

Fig2 based on the deviations of the datapoints from the fitted curve it appears that the WT GMP data in Fig2a, b, and c and Fig3a were previously published (fig2a, Parker 2017). This should be clearly indicated.

Fig2d appears to show transport rates. Rates are measured in amount of substrate transported per time unit per amount of protein. Not in "pmols transported".

P10,center buried surface area > specify the software used to calculate the area

P10,bottom stabilization due to oligomerisation, dimer formation in SDS-PAGE > stabilization of membrane proteins following membrane reconstitution is commonly observed, similar to oligomeric bands upon SDS-PAGE analysis of proteoliposomes. These are not established methods for providing indications or even hints on the oligomeric state, in contrast to proven methods such as Blue Native PAGE or glutaraldehyde cross-linking. The observations indicated do not have any relevance in addressing oligomeric states, nor provide initial indications. Remove to avoid

confusion.

Fig4+5 the concept of "lipid density" needs to be explained better. What is meant here? Residence time? How can high lipid densities be achieved, given that all lipids have similar volumes and masses.

P12, center the Y281A mutant is transport inactive. Is it similar to the K118A mutant still binding competent? If yes, how can the functional difference between the mutants concerning cooperativity be explained?

Fig5 legend reads DPPC, figure indicates DMPC

P17, top dynamic oligomerisation interface > what is meant with "dynamic"?

P20, top thermal stability > indicate excitation wavelength

Reviewer #3:

Remarks to the Author:

This paper by Parker et al. reports the structural, functional and computational characterizations of Vrg4, which is a yeast member of the nucleotide sugar transporters (NSTs). In eukaryotic cells, nucleotide sugars supply carbohydrates for glycosylation in the endoplasmic reticulum or Golgi apparatus, which are transported into these compartments by NSTs. The first structure of Vrg4 bound to GDP-mannose, reported by the authors' group, revealed the architecture of this transporter and defined the mechanism of nucleotide sugar recognition. In this paper, the authors further determined the GMP-bound structures of Vrg4 and found that GMP is bound to the substrate-binding site with quite different conformation from that of GDP-mannose. Although the resolution of the crystal structures reported in this paper, as well as the previous structure of GDP-mannose-bound form, are not high enough to discuss the atomic interactions and conformations of the bound ligand, the elaborated mutational analyses clearly corroborated the author's hypothesis about the substrate binding modes. In addition, the authors pursued the physiological role of the dimer formation of Vrg4 and the bound short chain lipids, using biochemical assay and computational simulations. The combination of structural, functional and computational analyses explained the structural basis of different specificities on counter-substrates and the lipid regulation — two important but poorly-understood aspects of the NSTs and other secondary active transporters.

Overall, this paper represents an interesting piece of work in the transporter biology, and therefore is recommend for publication. Meanwhile, I felt that several points in the conclusions are not convincingly supported by the presented data and literature, and thus need to be addressed before publication.

Major points:

- As mentioned in Introduction (line 14, page 3), the lipid-mediated protein regulation could be explained either by the specific interaction with lipid molecules or by the bulk/local properties of bilayer (such as thickness and fluidity). While this study addresses the former possibility in depth, the latter is not addressed. The authors previously reasoned that hydrophobic mismatch plays a role in the observed short-chain lipid dependency of Vrg4 (Parker et al., *Nature*, 2017), a hypothesis supported by the analysis of the TMD lengths of various proteins by Sharpe et al. (*Cell*, 2010). Could the authors provide any insights into the bilayer thickness from the MD simulations?

- The idea that DMPC binds to Vrg4 and activates the transporter sounds interesting, but it was not clear to me whether short-chain lipids are actually abundant in the ER/Golgi to enable such regulation, from the cited literature (ref. 23). Ref. 23 does mention that sphingolipids and cholesterol are abundant in the plasma membrane, resulting in thicker bilayer, but does not mention that specific short-chain lipids are abundant in the ER/Golgi. Therefore, there remains the question as to whether the DMPC-dependent conformational change of Vrg4 is biologically relevant in a cellular context. The conclusion would be more convincing if the authors could include literatures or discussions on the short-chain lipid contents of different membrane environments.

- The different binding modes between GMP and GDP-mannose is one of the important points in this paper; however, the referee still not convinced the author's interpretation of the density maps of the substrates. Is it impossible to model the GMP molecules in a similar conformation as that of GDP-mannose in the previous structure? The author should show and compare the shape of the density of GMP and GDP-mannose substrates, viewing exactly from the identical direction against the substrate binding sites. Also, the authors should show the omit maps of the transporter side chains that are discussed to be important for the substrate recognition.

Other important points:

- The authors conducted both the coarse-grained and all-atom MD simulations, but it is not clear the results are obtained from which simulation (except for the density plot in Fig. 4c). The authors should note about which results are obtained which simulations.

- We noticed that there are two recent publications on CMP-sialic acid transporters (Ahuja et al., eLife, 2019; Nji et al., NSMB, 2019). The authors should mention about these structures in Introduction. I also suggest adding structural or mechanistic comparison of Vrg4 and CST, which might help deepen the proposed mechanism of substrate recognition and obligatory exchange by NSTs. (DOI: 10.1038/s41594-019-0225-y and DOI: 10.7554/eLife.45221)

- Line 4, page 7. The discussion about Y281, S266 and S269 is confusing. The structures show that S266 and S269 interact with the guanine base, implying its importance for nucleobase recognition, but in the mutational assay, these residues showed different impacts on GMP and GDP-mannose transport, implying its impact on sugar recognition. Do the authors mean that nucleobase recognition is so significant for GMP transport that its affinity decreased upon mutation, but less significant for GDP-mannose transport because the sugar is recognized by other residues?

- Line 7, page 13. From the statement "When re-analyzing our simulation data for the short chain lipid, DMPC,", it is not clear whether the authors observed DMPC accumulation in the same DMPC/DPPC simulation, or they performed separate simulations in DMPC or DPPC. By reading the Methods section the authors have performed simulations only in the DMPC/DPPC mixed bilayer. The sentence should be revised so that it is clear DMPC accumulated in the additional site in the DPPC/DMPC simulation, while DPPC did not.

- Line 1, page 20, Methods. The authors show thermal stability assays in detergent micelles and liposomes to examine the stabilizing effects of lipids. However, as seen by the noisy derivative plot of the "Lipid" sample in Supplementary Figure 4, the protein melting behavior is significantly different from the detergent sample, and one should be especially careful interpreting such results from the microscale thermophoresis in a different setup. Ideally the authors could include a control experiment - (1) liposomes without proteins, (2) liposomes with pre-heated proteins, or (3) liposomes solubilized by detergents.

Minor points:

Line 2, page 11. The statement "Analysis either via mass spectrometry or SEC-MALS" lacks reference. Does it come from the authors' original data or from other published studies?

Line 6, page 14. Did the authors observe any lipid densities near TM9 in the previous and current structures?

Line 18, page 14. From reading ref. 10, the switching conformation of Vrg4 is only speculated from the domain swapping modelling and mutational assays. Therefore "we previously identified" seems too strong and the words like "suggested" or "proposed" might be more suitable here.

Line 14, page 9. Amine group  amino group

Line 8-10, Page 14. The importance of TM9 of TpT is also discussed in Lee et al. (Nat Plant 2017; DOI: 10.1038/s41477-017-0022-8), which should be cited here.

Line 3, page 21. Charmm36  CHARMM36

Response to reviewers' comments:

We wish to thank the three reviewers for taking the time to read and provide helpful critique of our manuscript. We have addressed the main points raised through additional experimental data, new figures and re-wording/clarification of the text. Below are specific comments to the points raised.

Reviewers' comments:

Reviewer #1 (Remarks to the Author):

Parker et al present a structure of the nucleotide sugar transporter Vrg4 in the presence of GMP, which they compare with a previously determined structure in the presence of GDP-mannose. They use the observed structural differences in the ligand binding modes to interpret differences in kinetics of transport of the two substrates. In addition, they analyze the oligomeric state of the protein (likely dimeric in lipid bilayers) and reevaluate previous MD simulations to propose a role for lipids in regulation of transport. The work advances our knowledge in the transport mechanism.

The work is technically well done, but I have some doubt about the authors' (over)interpretation of the data.

Specific points (in no particular order) Page4: There are 8 transporters in the unit cell, but only three contain GMP. Apparently, exactly the same three transporter molecules in each unit cell hold GMP. This must mean that there are structural differences between the transporters in the unit cell that force preferential binding to some, or disfavor binding to the others. This must be discussed, and explained. On a similar note: It is always Chain E that has the unusual binding pose of GMP. What is special/different about Chain E? A careful analysis of differences may help the prediction and simulation of subtle changes in the binding site, and the dynamics thereof. At least this should be carefully discussed.

It is unclear to us why only three of the eight molecules in the unit cell contain ligand. It is interesting to note that it was the same three molecules that contained the GDP-mannose ligand in our previous study. We should highlight here that to obtain the ligand co-crystal structures we had to initially set up the crystallisation experiment in the presence of 10mM GMP, and then additionally soak the crystals in 20mM GMP, which we state in the methods section. Why only three molecules then contain ligand is a mystery. However, we cannot see from the current structures or MD analysis any significant differences in these three molecules; we have looked at r.m.s.d, B-factor and stereochemical analysis and can see no significant differences. We think this is just a quirk of the crystal packing and the LCP set up. Subtle changes in the binding site are already discussed in the Results section of the paper.

Page 8 and discussion: It is unclear to me what exactly the authors are trying to say about the kinetics of transport. First, from their previous paper, I understand that GMP and M-GDP display similar IC50 (~Km) values (Fig 2 in the Nature paper) but different Vmax (?) (deduced from Figure 1e of the Nature paper). This would be consistent with lower binding affinity for GMP than for M-GDP. In vivo, in the lumen of the Golgi/ER M-GDP concentrations will be low, because M-GDP is converted by enzymes upon entry. Therefore, there will just be GMP left to drive the antiport. This has not much to do with kinetics, but just with concentrations. On the cytoplasmic side of the membrane, the higher concentration of M-GDP and very low concentration of GMP favors use of M-GDP as substrate. To what extent do the kinetic

differences (=differences in apparent affinities) play a role too? A k_{cat}/K_m determination could shed light.

Related: Fig 2d of the current manuscript shows that the kinetic differences between GMP and M-GDP are not consistent. The Grey and blue bars (3HGMP/GDP-M vs 3HGDP-M/GMP) are identical, which is difficult to reconcile with significant kinetic differences. This must be explained.

We interpret our data as supporting the hypothesis that Vrg4 is able to transport one substrate faster than the other, which could be a route to regulating transport in the presence of saturating concentrations of the different ligands in the cell. The structural and biochemical data provide a mechanistic explanation for this phenomenon. It may help to think about the process differently, i.e. that GDP-mannose enables Vrg4 to move faster because it interacts more effectively with the gating helices as it makes more interactions compared to GMP, which we show in this paper. We speculate this has a role physiologically, but as you point out, without knowing the concentrations of Vrg4 and ligands in the cell we can't know for sure the extent to which this phenomenon influences physiological rates of transport. However, we felt it important to highlight the very real possibility that it may play an important role. As to the comment on the grey and blue bars in Figure 2d, these represent identical experimental conditions as both sides of the liposomes have concentrations of ligand above the IC_{50} values. We have now included a toned-down version of our original statement.

Fig 4b: the SDS-PAGE interpretation seems cherry picking to me: On the one hand the authors claim that in detergent monomers are favored (MS and SEC-MALS), even in the presence of lipids, but in SDS (also a detergent, but now a denaturing one) they claim that dimers are observed. In my view SDS band in the presence of lipid could just be a big artefact of denatured protein (as has been observed many times in the past).

Agreed. We have now performed additional experiments to support this part of our study (that lipid is required to aid dimerization). Specifically, we have undertaken cross linking experiments on Vrg4 in the presence of additional lipid (yeast polar lipid) (See Supplementary Fig. 6). This new data shows that upon addition of lipid we observe an increase in the dimer species.

Page 12: the mixed dimer effects need more explanation. I am not convinced about the 50% vs 75% difference. The statistics is not explained (Notwithstanding the ticked box in the reporting summary). How many biological replicates were used? It could also be that the K118A mutant reconstitutes poorly, because it may not be stable in the absence of substrate, and that the ratio of reconstituted proteins is not the same as the ratio of purified proteins in detergent solution. In any case this result need much more discussion. The way it is presented now seems pure speculation and overinterpretation.

To support our earlier hypothesis we have performed additional experiments at 25:75 and 75:25 ratios (WT:K118A) (see Figure 4e and Supplementary Fig. 4). In addition, the methods have been expanded to explain the biological replicates. Our new data supports the previous analysis that Vrg4 operates as a monomer but does oligomerise in the lipid environment. We have included SDS-PAGE data that confirms the same amount of protein is present across all the ratios tested.

P15 “substantially different rotamer configuration” Without showing densities, I am not convinced given the moderate resolution.

We have now shown the 2Fo-Fc density maps for K289 in Supplementary Fig. 3, which show the different rotamer position of K289 in the structures. We have also included a superposition of the three rotamer positions as an additional panel.

Reviewer #2 (Remarks to the Author):

The manuscript of Parker et al. follows up on Parker & Newstead, 2017, that detailed the structure of yeast GDP-mannose transporter, Vrg4, in a conformation open towards the golgi lumen and without ligand and with GDP-mannose bound. The novel structure presented in this manuscript details the Vrg4 structure in the same conformation but in the presence of GMP. This is a relevant structure as Vrg4 exchanges cytoplasmic nucleotide sugars for nucleoside monophosphate from the golgi lumen. The authors show that GMP, surprisingly, has two major binding modes in the Vrg4 conformation open to the golgi lumen. In addition, the interaction of certain lipids with the transporter is characterized computationally and the functional relevance of a potential dimer is studied, leading to the suggestion that the monomers in the dimer functionally interact.

Though the structure of GMP-bound Vrg4 is novel and important, the additional sections of the manuscript on the characterization of the substrate binding, the relevance of lipids, and the functional interactions in the dimer are less convincing. The mutants indicated in Fig2 may contribute to discriminating between GMP and GDP-mannose, but these are very small at most, and negligible compared to the K289A mutant published previously (Parker 2017). The explanation on differences in transport rates is interesting, but speculative and confusing, given the large emphasis on substrate binding while the apparent affinities for both substrates are not different. The sections on the relevance of lipids are exclusively computationally and would benefit from functional studies to support the claims. The characterization of the dimer overall is interesting and relevant, but would require additional supporting evidence that this dimer truly exists.

The manuscript of Parker et al. details several relevant and interesting topics on nucleotide sugar transporters. I find the current version of the manuscript a bit premature. The manuscript would benefit from a more clear description of the experiments and a clearer interpretation of the data. In addition, more extensive or additional experiments (see below) are required to make the claims stronger.

Concerning the functional and mechanistic interpretations of substrate binding I have a few concerns:

1) The functional experiments are insufficiently detailed. For every figure, it should be made clear what substrate is in what compartment and at what concentration. The current poor descriptions leave unwanted room for different interpretations.

We have added more information to the methods to clarify further the concentrations used for each assay.

2) The IC50 values measured in Fig2 and Fig3 differ only 2-4 fold. This is generally not considered as significant and within the range of experimental error. The authors should detail

why this small difference is taken as relevant. Error bars should be added to every datapoint in all IC₅₀ curves.

We would respectfully disagree with this point. A 2-4 fold difference in IC₅₀ value is significant when the errors in the measurements are small, as is the case here. Each experiment was repeated fully in triplicate using independently made stocks of protein and substrate. Additionally, different concentrations of substrates are also used to make sure the calculated IC₅₀ value is accurate. The mean value shown is the mean of every IC₅₀ calculated and not the error of each concentration used. This is why errors on the representative graph are not shown but are for the calculated IC₅₀ values. For complete transparency we have now shown all the graphs for each individual experiment in new Supplementary Figures, 9 & 10.

3) Control the direction (P8) > what is the evidence that Vrg4 actually controls the direction? An alternative and more straightforward interpretation is that the enzymes in the golgi and the cytoplasm contribute to high concentrations of GMP in the Golgi and GDP-mannose in the cytoplasm. These opposite concentration gradients may already suffice to “control the direction”.

We have reworded the text as follows “An important question has been how these systems ensure the direction of transport for their ligands.”

4) On P8 the authors compare the transport rate between GMP and GDP-mannose to conclude that GDP-mannose is transported more efficiently. The efficiency of transport is best reflected in the V_{max}, which requires that measurements are done at saturating substrate concentrations. Was this the case for the data presented in Fig2d?

Yes this was the case. These experiments were performed at saturating substrate concentrations (0.1 mM for GMP and GDP-mannose and 0.5 mM for AMP) this is detailed in the methods.

5) P14, bottom. The direction of transport / different rates of transport > see above. All transport measurements are done in proteoliposomes. Given the reconstitution method, it appears likely that the transporter is reconstituted in equal amounts in two different orientations. That implies that potential differences in substrate affinities on the cytoplasmic and golgi-luminal side could not be measured (this is also not claimed by the authors).

We don't think this is an issue as the experiments are carried out under saturating conditions and we can assay the movement of both ligands in either direction. If differences in substrate affinities do exist (which we think unlikely given the similar IC₅₀ values calculated in our previous study) these would not contribute to the difference in rates we observe.

As information on the asymmetry of the substrate binding sites in combination with the physiological concentrations of GMP and GDP-mannose in either compartment is absolutely required to allow any prediction on “directionality”, the discussion on the different rates is irrelevant. An increased transport rate of GDP-mannose alone may support increased export from the golgi-lumen as well. Revise to indicate the modest role of the rate with respect to the action of catabolytic and anabolytic enzymes and potential asymmetry in the substrate affinities on cytoplasmic and luminal binding sites.

We have revised the text to downplay the role and replaced with “may play”.

6) P8+P15 the more dynamic GMP is simply less efficient at docking into the binding site > what is the evidence that GMP is less efficient at docking? Though the structures indicate multiple binding modes, the IC50, which is an indirect indicator for affinity, is identical for GMP and GDP-mannose. How do the authors reach the conclusion that the “recognition of GMP is less efficient than the GDP-mannose”?

Our conclusion is based on the structural data, which shows that GMP makes far fewer interactions with the protein. We agree that in our previous version of the manuscript our use of the word ‘efficient’ was open to different interpretations. We have now clarified this point to indicate that our data shows that GMP is more dynamic in the transporter, making fewer interactions. We suggest it is this property of the molecule that is responsible for the slower rates of transport observed compared to GDP-mannose.

Concerning the study on the oligomeric state and lipid-dependence of Vrg4 I have the following concerns:

7) The evidence that the protein is oligomeric in the membrane is exclusively based on the available structures that show dimers (next to monomers) in an orientation that physiologically could make sense.

The evidence that the protein must oligomerise within a lipid environment is also based on the dominant negative experiments, which we think are very compelling. We have now expanded these experiments to include further ratios which supports our original hypothesis that the protein must form oligomers (Fig 4e).

However, the dimer may be an artefact from the high protein concentrations and the 2D plane resulting from the LCP. Additional independent proof that Vrg4 forms dimers in membranes is absolutely required. As the tentative structure of the dimer is known, I suggest to perform cysteine cross-linking experiments (oxidative or using cross-linkers) for this purpose.

We agree and have now performed additional experiments in the form of glutaraldehyde crosslinking the supports our claim that lipid aids dimer formation of Vrg4 (SI Fig 6).

8) P11, is the protein active in bilayers composed of only DPPC and DMPC that were used for the MD?

This is unknown, due to the high phase transition temperature of DPPC (41°C), which precludes its use in liposome assays. However, we carefully designed our MD experiments to provide the most direct measure possible of lipid chain length on Vrg4 binding. The difference between the chosen lipids in Martini is simply one particle, thus any differences in headgroup chemistry and chain saturation will not complicate interpretation of the data. Most other mixtures of lipids would not have provided such a clear analysis. To ensure that the lipids were all within the appropriate phase, the simulations were run at 323 K, in keeping with standard practice for simulations using the Martini force field. Using these conditions would be difficult to achieve experimentally, and might lead to unforeseen effects on the protein. It is very common practice to combine MD simulations using a model membrane for clarity of analysis with experimental analyses which use lipids carefully chosen to maximise

protein stability and activity. Therefore, we believe that the comparison made in the manuscript is sound.

9) P12, the studies on the functional significance of a potential dimerization is interesting, but not sufficiently convincing. Membrane reconstitution always leads to some experimental variation, the amount of functionally reconstituted protein will vary. In the setup chosen by the authors, the difference between cooperative and independent functioning essentially depends on the activity measured for one ratio. This is too little. Additional intermediate ratio's should be tested to make this result reliable.

We have now included more ratios which support our original hypothesis (Figure 4e).

10) P12, bottom. Stabilizing effect > it is not clear what is meant. The reference refers to a transporter that appears to use a very different transport mechanism than Vrg4. Clarify.

We have expanded and clarified this part of the manuscript to detail the implications of the mixed dimer and our interpretation of these results.

11) P13, top. In the preceding paper (Parker 2017) the effect of DMPC appears to be attributed to hydrophobic mismatch. In the current manuscript this effect appears assigned to a short chain lipid dependence, which is much more specific than hydrophobic mismatch. What is the basis for this novel interpretation of the data?

12) The observation of a stable interaction between DMPC and the protein in MD simulation is interesting. However, the mechanistic interpretation is far too early. If the authors wish to substantiate this claim, mutants should be made to disrupt this interaction followed by functional characterization. Alternatively, this section should be left out.

Response to points 11 & 12: Our evidence is based on the MD simulations, where we observe a specific lipid cavity that could only accommodate a short chain lipid. Whilst we agree with the referee that we cannot rule out the short chain lipid requirement is solely to do with hydrophobic mismatch, we believe the presence of this short chain lipid cavity was interesting, provocative and worthy of discussion within the paper. However, as we cannot ascertain which of these reasons explains the short chain lipid requirement we have toned down the text to reflect this.

Additional comments

P4, line2 kinetic difference > specify whether this is K_m or V_{max} or both
sfig1 place the proteins in the three panels in the identical orientation to demonstrate the different ligand orientations.

We have specified V_{max} but prefer to keep the orientations as in the original figure, as we feel this illustrates the GMP ligand more clearly.

P5, bottom what is the rmsd between the GMP and the GDP-mannose structures?

The r.m.s.d between the chains in the crystal structures is ~ 0.476 Ångstrom. This has been added.

P6, bottom affinity of this variant for GDP-mannose remains the same as wildtype > Fig2a does not depict any data for wt. Is this value based on literature?

Fig2 based on the deviations of the datapoints from the fitted curve it appears that the WT GMP data in Fig2a, b, and c and Fig3a were previously published (fig2a, Parker 2017). This should be clearly indicated.

Yes, this has been referenced (reference 10) and included in the methods.

Fig2d appears to show transport rates. Rates are measured in amount of substrate transported per time unit per amount of protein. Not in “pmols transported”.

Yes, this has been changed.

P10,center buried surface area > specify the software used to calculate the area
P10,bottom stabilization due to oligomerisation, dimer formation in SDS-PAGE > stabilization of membrane proteins following membrane reconstitution is commonly observed, similar to oligomeric bands upon SDS-PAGE analysis of proteoliposomes. These are not established methods for providing indications or even hints on the oligomeric state, in contrast to proven methods such as Blue Native PAGE or glutaraldehyde cross-linking. The observations indicated do not have any relevance in addressing oligomeric states, nor provide initial indications. Remove to avoid confusion.

We have now undertaken and included the glutaraldehyde cross linking experiments.

Fig4+5 the concept of “lipid density” needs to be explained better. What is meant here? Residence time? How can high lipid densities be achieved, given that all lipids have similar volumes and masses.

The data are weighted particle densities, achieved by replacing each particle with a normalised Gaussian. A higher density refers to a high number of lipid densities overlapping in a specific grid position over the course of the simulation - i.e. a lipid is in that exact position across many different timeframes.

To clarify we have added "densities calculated with the VMD volmap plugin" to the figure legend.

P12,center the Y281A mutant is transport inactive. Is it similar to the K118A mutant still binding competent? If yes, how can the functional difference between the mutants concerning cooperativity be explained?

We have included new Prometheus data that shows Y281A does not respond to ligand, suggesting this variant is unable to bind ligand.

Fig5 legend reads DPPC, figure indicates DMPC

Fixed, thank you.

P17, top dynamic oligomerisation interface > what is meant with “dynamic”?

We mean this to be that the protein is not always in a dimer configuration. I.e. during purification Vrg4 is monomeric.

P20, top thermal stability > indicate excitation wavelength

Done – 280nm.

Reviewer #3 (Remarks to the Author):

This paper by Parker et al. reports the structural, functional and computational characterizations of Vrg4, which is a yeast member of the nucleotide sugar transporters (NSTs). In eukaryotic cells, nucleotide sugars supply carbohydrates for glycosylation in the endoplasmic reticulum or Golgi apparatus, which are transported into these compartments by NSTs. The first structure of Vrg4 bound to GDP-mannose, reported by the authors' group, revealed the architecture of this transporter and defined the mechanism of nucleotide sugar recognition. In this paper, the authors further determined the GMP-bound structures of Vrg4 and found that GMP is bound to the substrate-binding site with quite different conformation from that of GDP-mannose. Although the resolution of the crystal structures reported in this paper, as well as the previous structure of GDP-mannose-bound form, are not high enough to discuss the atomic interactions and conformations of the bound ligand, the elaborated mutational analyses clearly corroborated the author's hypothesis about the substrate binding modes. In addition, the authors pursued the physiological role of the dimer formation of Vrg4 and the bound short chain lipids, using biochemical assay and computational simulations. The combination of structural, functional and computational analyses explained the structural basis of different specificities on counter-substrates and the lipid regulation — two important but poorly-understood aspects of the NSTs and other secondary active transporters.

Overall, this paper represents an interesting piece of work in the transporter biology, and therefore is recommend for publication. Meanwhile, I felt that several points in the conclusions are not convincingly supported by the presented data and literature, and thus need to be addressed before publication.

Major points:

- As mentioned in Introduction (line 14, page 3), the lipid-mediated protein regulation could be explained either by the specific interaction with lipid molecules or by the bulk/local properties of bilayer (such as thickness and fluidity). While this study addresses the former possibility in depth, the latter is not addressed. The authors previously reasoned that hydrophobic mismatch plays a role in the observed short-chain lipid dependency of Vrg4 (Parker et al., Nature, 2017), a hypothesis supported by the analysis of the TMD lengths of various proteins by Sharpe et al. (Cell, 2010). Could the authors provide any insights into the bilayer thickness from the MD simulations?

We have now included new data showing MD simulations in DMPC vs. DPPC membranes and show that Vrg4 is able to reduce the thickness of a DPPC bilayer in close proximity to the protein. We have included this in Supplementary Fig. 7.

- The idea that DMPC binds to Vrg4 and activates the transporter sounds interesting, but it was not clear to me whether short-chain lipids are actually abundant in the ER/Golgi to enable such regulation, from the cited literature (ref. 23). Ref. 23 does mention that sphingolipids and cholesterols are abundant in the plasma membrane, resulting in thicker bilayer, but does not mention that specific short-chain lipids are abundant in the ER/Golgi. Therefore, there remains the question as to whether the DMPC-dependent conformational change of Vrg4 is biologically relevant in a cellular context. The conclusion would be more convincing if the authors could include literatures or discussions on the short-chain lipid contents of different membrane environments.

Following this suggestion we did an exhaustive search of the literature but could not find any information on lipid length with respect to organelles. This is almost certainly due to technical limitations with mass spec, which may change in the near future. We accept this point is fair, but, importantly our data show that lipid length is important for Vrg4 function and this paper in particular provides a mechanism for this influence. It thus provides the platform from which more cell-based studies can be conducted and the in vivo influence ascertained. We have now removed the statement that short chain lipids are concentrated in the ER/Golgi.

- The different binding modes between GMP and GDP-mannose is one of the important points in this paper; however, the referee still not convinced the author's interpretation of the density maps of the substrates. Is it impossible to model the GMP molecules in a similar conformation as that of GDP-mannose in the previous structure?

In short, yes, it is not possible to model the GMP ligand in the same position as that of GDP-mannose. The OMIT difference maps shown in Supplementary Figure 1 clearly showed the GMP ligand in a different position relative to the GDP-mannose structure. The superposition of the GMP ligand positions with that GDP-mannose is shown in Supplementary Fig. 2. However, the positions are not radically different. In Supplementary Fig. 2c for example, you can see in chain E that the nucleotide base adopts a similar position to that in GDP-mannose, whereas in chains C and D the maps strongly suggest the base is flipped. However, notice that the alpha phosphate of the GMP in both chains is not that far from the beta phosphate in GDP-mannose. Thus, the overall recognition pattern is similar, but different enough to elicit the transport effects we observe.

The author should show and compare the shape of the density of GMP and GDP-mannose substrates, viewing exactly from the identical direction against the substrate binding sites. Also, the authors should show the omit maps of the transporter side chains that are discussed to be important for the substrate recognition.

The overlays of the GDP-mannose from our previous study (ref. 10) and GMP ligands (this study) are shown in Supplementary Fig. 2. We think this figure is enough to show the similarities and differences between the binding positions for GMP and GDP-mannose. Given that the side chain positions are very similar to those from the previously reported GDP-mannose structures we don't think it necessary to show the OMIT maps for these atoms. However, we do agree that the maps for K289 are important, and have now shown these in a new Supplementary Fig. 3.

Other important points:

- The authors conducted both the coarse-grained and all-atom MD simulations, but it is not clear the results are obtained from which simulation (except for the density plot in Fig. 4c). The authors should note about which results are obtained which simulations.

We have now clarified this in the text.

- We noticed that there are two recent publications on CMP-sialic acid transporters (Ahuja et al., eLife, 2019; Nji et al., NSMB, 2019). The authors should mention about these structures in Introduction. I also suggest adding structural or mechanistic comparison of Vrg4 and CST,

which might help deepen the proposed mechanism of substrate recognition and obligatory exchange by NSTs. (DOI: 10.1038/s41594-019-0225-y and DOI: 10.7554/eLife.45221).

We have now included these references in the introduction. However, we feel a more in-depth analysis is more suited to a review, especially given the different ligand positions within the two CMP-sialic papers.

- Line 4, page 7. The discussion about Y281, S266 and S269 is confusing. The structures show that S266 and S269 interact with the guanine base, implying its importance for nucleobase recognition, but in the mutational assay, these residues showed different impacts on GMP and GDP-mannose transport, implying its impact on sugar recognition. Do the authors mean that nucleobase recognition is so significant for GMP transport that its affinity decreased upon mutation, but less significant for GDP-mannose transport because the sugar is recognized by other residues?

We have now clarified this in the text.

- Line 7, page 13. From the statement “When re-analyzing our simulation data for the short chain lipid, DMPC,”, it is not clear whether the authors observed DMPC accumulation in the same DMPC/DPPC simulation, or they performed separate simulations in DMPC or DPPC. By reading the Methods section the authors have performed simulations only in the DMPC/DPPC mixed bilayer. The sentence should be revised so that it is clear DMPC accumulated in the additional site in the DPPC/DMPC simulation, while DPPC did not.

We have now clarified this in the text.

- Line 1, page 20, Methods. The authors show thermal stability assays in detergent micelles and liposomes to examine the stabilizing effects of lipids. However, as seen by the noisy derivative plot of the “Lipid” sample in Supplementary Figure 4, the protein melting behavior is significantly different from the detergent sample, and one should be especially careful interpreting such results from the microscale thermophoresis in a different setup. Ideally the authors could include a control experiment - (1) liposomes without proteins, (2) liposomes with pre-heated proteins, or (3) liposomes solubilized by detergents.

We have now included liposome solubilised in detergent. Liposomes without protein did not give a signal, data which is now included and liposomes with pre-heated proteins did not reconstitute.

Minor points:

Line2, page 11. The statement “Analysis either via mass spectrometry or SEC-MALS” lacks reference. Does it come from the authors’ original data or from other published studies?

This came from unpublished work during the course of this study. We have now removed this line from the manuscript to avoid confusion.

Line 6, page 14. Did the authors observe any lipid densities near TM9 in the previous and current structures?

We did not.

Line 18, page 14. From reading ref. 10, the switching conformation of Vrg4 is only speculated from the domain swapping modelling and mutational assays. Therefore “we previously identified” seems too strong and the words like “suggested” or “proposed” might be more suitable here.

Changed to ‘suggested’, thank you.

Line 14, page 9. Amine group  amino group

Changed.

Line 8-10, Page 14. The importance of TM9 of TpT is also discussed in Lee et al. (Nat Plant 2017; DOI: 10.1038/s41477-017-0022-8), which should be cited here.

We already also added this reference to the following sentence in the paper. “TM9 was also identified as being important for the transition between luminal and cytoplasmic facing conformations of the plant triose phosphate antiporter, TpT, a distant homologue of the NST family²⁶.”

Line 3, page 21. Charmm36  CHARMM36

Changed, thank you.

Reviewers' Comments:

Reviewer #1:

Remarks to the Author:

The authors have adequately addressed my previous concerns.

Reviewer #2:

Remarks to the Author:

The revised manuscript of Parker et al. addresses most of my previous comments satisfactory and still details several relevant and interesting topics on nucleotide sugar transporters. The inclusion of basic experimental details makes it now possible to appreciate the experimental outcomes. I do note that several of the figure legends are still incomplete or overly minimalistic.

I have the following remaining points.

I respectfully disagree with the argument of the authors on the significance of the IC₅₀ value. The authors argue that the error of the three independently determined IC₅₀ values suffices to indicate the quality of the data. However, this line of argumentation does not appreciate that the individual IC₅₀ values itself are derived values that are based the quality of the individual datapoints, which I strongly feel should be taken into account as well. In my opinion, the currently used analysis methods does not give a fair representation of the actual experimental error.

This professional disagreement may reflect personal styles in data analysis. As the authors provide the original data in the supplement, and possibly even the raw data as source-data-file, sufficient transparency is provided to allow expert readers of assessing this data themselves. I therefor conclude that this point has been addressed satisfactorily.

Glutaraldehyde crosslinking (S Fig 6c): the legend should explain the difference between + and ++, the reader should not have to dig into the methods section first. The methodological approach (mix purified protein with liposomes) is somewhat unusual as no concentrations of protein, detergent, and lipids are specified. This approach only makes sense if the detergent concentration is sufficiently high so that the liposomes are solubilized, else the lipid may scavenge all free detergent leading to aggregation of the protein. Please detail.

The observation of dimers upon SDS-PAGE of proteoliposomes cannot be taken as an argument in favor of dimer formation in the presence of lipids. The latter appears (see above) to have been demonstrated by the crosslinking already. SDS-PAGE is not an established method to demonstrate oligomeric states of membrane proteins. If the authors think otherwise, I encourage them to refer to manuscripts detailing and verifying this methodological approach, or generate such data themselves. Alternatively, this section is removed.

Functional role dimerization. Yes, the authors have included additional data points, but only for the lower panel. Thus for mixed dimers of K118A and WT a comparably high activity is found, suggesting a functional relevance of the dimer. For the upper panel on Y281A no additional datapoints are added, implying that my initial concerns on the data are not addressed. I suggest this data is either left out or extended with two additional ratios similarly as the lower panel.

Page 12, bottom: "greater than 50 % reduction". Cooperative transport should lead to either a negative or a positive deviation from the 50 % value, not only a reduction as stated in the text.

Reviewer #3:

Remarks to the Author:

I think the authors adequately addressed the concerns raised, including the additional results. The paper now seems to be suitable for the publication in Nature Comm.

Response to reviewers' comments:

We wish to thank again the three reviewers for taking the time to read and provide helpful critique of our manuscript. We have addressed the additional points as specified below:

Reviewers' comments:

“Glutaraldehyde crosslinking (S Fig 6c): the legend should explain the difference between + and ++, the reader should not have to dig into the methods section first.”

We have added this information to the figure legend.

The observation of dimers upon SDS-PAGE of proteoliposomes cannot be taken as an argument in favor of dimer formation in the presence of lipids. The latter appears (see above) to have been demonstrated by the crosslinking already. SDS-PAGE is not an established method to demonstrate oligomeric states of membrane proteins. If the authors think otherwise, I encourage them to refer to manuscripts detailing and verifying this methodological approach, or generate such data themselves. Alternatively, this section is removed.

We have added the following text to reduce the strength of our original statement. “We also observe possible dimer formation in SDS PAGE analysis of Vrg4 reconstituted into liposomes, which is not seen when the protein is in detergent, even at high concentrations of protein”

Page 12, bottom: “greater than 50 % reduction”. Cooperative transport should lead to either a negative or a positive deviation from the 50 % value, not only a reduction as stated in the text.

We have amended the text as follows, “If Vrg4 is a functional dimer we would expect to see a greater than 50 % reduction in transport.”